

# Biomes of the world under climate change scenarios: increasing aridity and higher temperatures lead to significant shifts in natural vegetation

Carmelo Bonannella[1,2], Tomislav Hengl[2], Leandro Parente[2] and Sytze de Bruin[1]

[1] Laboratory of Geo-Information Science and Remote Sensing, Wageningen University and Research, Wageningen, Netherlands
[2] OpenGeoHub Foundation, Wageningen, Netherlands

## ABSTRACT

The global potential distribution of biomes (natural vegetation) was modelled using 8,959 training points from the BIOME 6000 dataset and a stack of 72 environmental covariates representing terrain and the current climatic conditions based on historical long term averages (1979–2013). An ensemble machine learning model based on stacked regularization was used, with multinomial logistic regression as the meta-learner and spatial blocking (100 km) to deal with spatial autocorrelation of the training points. Results of spatial cross-validation for the BIOME 6000 classes show an overall accuracy of 0.67 and $R^2_{logloss}$ of 0.61, with "tropical evergreen broadleaf forest" being the class with highest gain in predictive performances ($R^2_{logloss} = 0.74$) and "prostrate dwarf shrub tundra" the class with the lowest ($R^2_{logloss} = -0.09$) compared to the baseline. Temperature-related covariates were the most important predictors, with the mean diurnal range (BIO2) being shared by all the base-learners (*i.e.*,random forest, gradient boosted trees and generalized linear models). The model was next used to predict the distribution of future biomes for the periods 2040–2060 and 2061–2080 under three climate change scenarios (RCP 2.6, 4.5 and 8.5). Comparisons of predictions for the three epochs (present, 2040–2060 and 2061–2080) show that increasing aridity and higher temperatures will likely result in significant shifts in natural vegetation in the tropical area (shifts from tropical forests to savannas up to $1.7 \times 10^5$ km² by 2080) and around the Arctic Circle (shifts from tundra to boreal forests up to $2.4 \times 10^5$ km² by 2080). Projected global maps at 1 km spatial resolution are provided as probability and hard classes maps for BIOME 6000 classes and as hard classes maps for the IUCN classes (six aggregated classes). Uncertainty maps (prediction error) are also provided and should be used for careful interpretation of the future projections.

# INTRODUCTION

Climate change is one of the biggest threats to human civilization, with slowly accumulating effects and unknown instabilities in front of us and future generations. To assess the potential impacts of climate change on the environment and to help us mitigate and

Corresponding author
Carmelo Bonannella,
carmelo.bonannella@opengeohub.org

prepare for negative effects, scientists offer predictions of possible futures including global maps of the Earth's environment in the future (*Dow & Downing, 2016*). Global datasets projecting the state of the Earth's environment include future climate predictions *e.g.* Representative Concentration Pathways (RCPs) (*Hayhoe et al., 2017*), future land use predictions (*e.g.*, *Chen et al. (2020)* and *Hurtt et al. (2020)*), human population scenarios (*Jones & ONeill, 2016*), future terrestrial ecosystems maps (*Nolan et al., 2018*), future ecosystem productivity (*Yin et al., 2023*), and future gridded emissions (*Fujimori et al., 2018*). Even though the accuracy of these projections in the far future cannot currently be validated, such exercises are deemed useful as they help reveal patterns and assess the impact of scenarios. In essence, there are two main approaches to envision the future state of Earth's environment (*Hayhoe et al., 2017*; *Reichstein et al., 2019*):

1. Process-based mechanistic modeling: simulating evolution of the environment using biophysical process-based Earth System Models (ESM);
2. Data-based modeling: training predictive models using observations from the past and then extrapolating these models into the future;

Process-based modeling is often preferred by physicists as the relationships between model entities are explicitly defined. Examples of projected changes of land use based on the global Earth System Models are the LUH2 project (*Hurtt et al., 2020*) and the Lund–Potsdam–Jena managed Land (LPJmL) model (*Rolinski et al., 2018*). In the case of data-based modeling, predictions and results of analyses are based on finding relationships between the target property and covariates and then fitting statistical models that are next used to predict values based on unseen combinations of states in feature space. Two common approaches here are: (1) use actual ground observations *i.e.* monitoring stations to fit spatiotemporal models (*Hengl et al., 2018*), and (2) use complete Earth observation data cubes and then basically all pixel combinations to visualize and model relationships (*Mahecha et al., 2020*). An advantage of the data-based modeling is that it is often computationally less demanding than process-based modeling and it can be extended by adding more covariates (*Beigaite et al., 2022*). In addition, process-based modeling requires several assumptions and, in the case of chaotic behaviour or non-linear spatial scaling of features, it is often difficult to produce credible predictions. On the other hand, data-based modeling comes with the risks of producing poor predictions in the extrapolation space and the models are often difficult to interpret (*Meyer & Pebesma, 2021*). Yet, strict data-based modeling requires neither subjective parametrization nor model assumptions, and hence it can be considered less complex to start with. It is not to say that the approaches are mutually exclusive and cannot be combined: there is a full spectrum of models from process-based to data-based, which includes hybrid physics-based data-driven models. Different approaches exist in this sense: using data-driven models but constrain the results with boundary conditions derived from physics-based climate models (as suggested by *Lindgren et al. (2021)*), including the representation of natural processes in the data-driven model (*Higgins et al., 2012*) or using process-based models whose results have been parametrized and calibrated on real data (*Higgins, Conradi & Muhoko, 2023*).

Predictions of future states of climate, land cover, terrestrial ecosystems, human population and similar have proven to be useful, with many of the datasets being frequently

cited and used to communicate our possible futures (https://probablefutures.org/). *Su, Gabrielle & Makowski (2021)* modeled yield gains under Conservation Agriculture (CA) and various practices for the future climate scenarios and found out that overall performance of CA will most likely decrease in the future in most temperate regions in South America, including Uruguay, southern Brazil and northern Argentina for barley, cotton, rice, sorghum and sunflower. *Krause et al. (2022)* has recently modelled the impacts of anthropogenic land cover changes on global gross primary productivity (GPP) using maps of historical agricultural expansion and future land-use changes based on the 25 km resolution LUH2 dataset (*Hurtt et al., 2020*). Their results indicate that global GPP might get further reduced owing to agricultural expansion and to extents that depend on the prevailing scenario. *Beigaite et al. (2022)* provides predictions of future distribution of MODIS vegetation types using machine learning and focusing on climate extremes (*e.g.* extreme cold days). Their results indicate that prediction accuracy can be improved by extending the averaged climatic conditions with maps of climate extremes *e.g.* bioclimatic variables and similar.

When it comes to mapping future vegetation, only a few datasets are available and typically at coarse resolutions. *Nolan et al. (2018)* provides predictions of terrestrial ecosystems in the future as a function of annual temperature and simple logistic spline regression with ordered categories. Their results suggest that terrestrial ecosystems are at risk of major transformation. Despite these recent efforts, there is still no analysis of the main future trends in air temperature and precipitation and the magnitude of such change on potential vegetation on a global scale. Furthermore, most of these datasets are provided without per pixel uncertainty estimates. The existence of various biome classification schemes makes things even more confusing, since they can be overly subjective (*Higgins, Buitenwerf & Moncrieff, 2016*) and in some cases they implicitly invoke climate (*Moncrieff, Hickler & Higgins, 2015*) in their definition: many of the early biome classification schemes included climate in their definition as a proxy for functional characteristics, traits and adaptations that were difficult to map properly at a global scale (*Moncrieff, Bond & Higgins, 2016*) and only later on schemes based on plant functional traits (PFTs) or ecosystem productivity have been developed; a paradygm shift has also taken place in the last decades, from considering biomes a deterministic entity to a more dynamic concept, a result of an ensemble of different processes and feedback loops (*Mucina, 2019*). However, the lack of datasets at high resolution that could be used to predict biome envelopes that follow the functional-based classification scheme is a limitation for its application to a global scale. Scientific studies that use data-driven approaches to forecast the state of vegetation into the future are usually limited on the spatial scale, spanning one or more countries or one continent at most (*Zevallos & Lavado-Casimiro, 2022*; *Maksic et al., 2022*), while another limitation consists in the usage of mostly one algorithm only (Random Forest) to conduct the analysis. The purpose of this study is to use a data-driven approach to provide consistent projections of future potential natural vegetation under different climate scenarios, including uncertainty estimates: we provide projections of 20 biomes for three (3) climatic scenarios (RCP 2.6, 4.5 and 8.5) for the future 60 years. To do that, we extend the work of *Hengl et al. (2018)*, which used a biome classification scheme based

on PFTs and tried to spatialize it to the whole globe by using an ensemble of climatic, topographic and remotely-sensed predictor variables. Compared to *Hengl et al. (2018)*, we apply the following three substantial improvements:

1. Instead of only using Random Forest, we use an ensemble of three learners of different types, which allowed quantifying the prediction uncertainty;
2. For each pixel we provide class probabilities and prediction errors computed by bootstrapping;
3. Modeling is done using a consistent set of covariates so that the effects of climate change are controlled purely by the climatic projections.

The paper is divided in four parts: (1) we first describe our predictive mapping framework based on using biome training points (*Harrison, 2017*); (2) we evaluate the accuracy of the fitted ensemble model using spatial cross-validation and generate predictions for the three future scenarios; (3) we next aggregate predictions according to the IUCN Global Ecosystem Typology classification system (*Keith et al., 2020*) to make our product comparable with an international standard and (4) we finally highlight the most pronounced changes per continent and biome type.

## MATERIALS AND METHODS

### General workflow

We modeled the potential distribution of biomes on a global scale for current and future time periods using an ensemble machine learning approach. The model was trained on reference biome data compiled from pollen and fossil reconstructions (*Harrison, 2017*) along with regional environmental variables describing topography and long-term climatic averages. We used CHELSA climatological data (*Karger et al., 2017*) from the time period 1979–2013 to simulate the baseline potential natural distribution of biomes for the current (2022–2023) time period: since our goal was to model the potential natural vegetation, we tried to predict which PFT-based class of biome would be the dominant one in a specific location based on environmental variables only. Future climatic conditions instead cover the epochs 2041–2060 and 2061–2080. For the future epochs we considered three different climate change scenarios using the concept of "*Representative Concentration Pathways*" (*Van Vuuren et al., 2011*), or, in short, RCPs. The ones used in this study are RCP 2.6, RCP 4.5 and RCP 8.5.

The output of the projections is provided as probability maps (0–100%) at 1 km spatial resolution, with the probabilities in each pixel summing to 100%. For each class we also provide model uncertainty maps. We excluded the continent of Antarctica, because of the presence of permanent ice areas and lack of training points. Also, other areas covered by water bodies, barren land and permanent ice according to ESA's global land cover maps for the period 2000–2015 (*ESA, 2017*) were excluded from the analysis. We generalized the 20 biome classes analyzed in this study to six classes following the Global Ecosystem Typology classification system employed by the International Union for Conservation of Nature (IUCN) (*Keith et al., 2020*). We then compared the two epochs for each of the climatic scenarios with the current time period: using the latter as a baseline for the distribution

of potential natural vegetation, the goal was to identify those areas where the change in climatological conditions could lead to a shift in the potential distribution.

## Training points

We used the BIOME 6000 data set, compiled by *Harrison (2017)*, with additional 350 pseudo observations to cover under-represented areas in South America (Fig. 1) as described in *Hengl et al. (2018)*, for a total of 8,959 points. The BIOME 6000 project aims to reconstruct past vegetation distributions from pollen and fossil records from different time periods, from the recent past (the last 50 years) to approximately 21 ka ago; in this study, following *Hengl et al. (2018)*, we only used the points belonging to the most recent time period. The method, described by *Prentice & Webb III (1998)*, was used to assign the recovered *taxa* to PFTs, which were next ascribed to a specific biome following PFT-based biomes definitions. From the first version of the data set to its final publication by *Harrison (2017)*, almost 20 years have passed: over this period, multiple surveys have been conducted on the same locations, resulting in more than one biome reconstruction per location. Furthermore, initially absent regions have been added to the original data set. To avoid issues with harmonization of nomenclature between biomes, *Harrison (2017)* provide a standardized classification legend that can be globally applied (32 biomes in total) and a *megabiome* classification legend (8 megabiomes in total). While the *megabiome* system implies a necessary loss of information due to generalization, the original standardized classification system devised by *Harrison (2017)* has been considered too detailed and location-specific to be used for global modeling (*Hengl et al., 2018*). We adopted the 20 classes (Fig. 2) system devised by *Hengl et al. (2018)* for the sake of data-model comparison.

## Predictor variables

A total of 72 spatially explicit and harmonized variables representing climatic, bioclimatic and topographic factors were used for modeling purposes. All the layers were resampled to a standard grid covering latitudes between 87.37°N and 62.0°S and reprojected to the coordinate reference system EPSG:4326 before the analysis. The original spatial resolution of the layers was used during the spatial overlay with the point dataset, while for the rest of the calculations all the layers were resampled to a spatial resolution of 30 arcseconds (approximately 1 km at the equator).

We used long-term climate data and projections as provided by the CHELSA project (*Karger et al., 2017*). For future scenarios, we followed the work of the Intergovernmental Panel on Climate Change (IPCC) Assessment Reports (AR) based on narratives and outcomes of the Coupled Model Intercomparison Project (CMIP). IPCC AR5 (*IPCC, 2014*) featured CMIP5 model results using the concept of representative concentration pathway (RCP), where each projected climatic scenario is labelled according to a possible increase in radiative forcing (from 2.6 to 8.5 $W/m^2$) values by 2100 due to increase in greenhouse gasses (GHG) emissions. The new IPCC AR6 (*IPCC, 2021*) featured instead CMIP6 model results while using a different concept, the "Shared Socioeconomic Pathways" (SSP): while RCPs did not include any socioeconomic factors in their modelization, SSPs included several assumptions on how population growth, technological development,

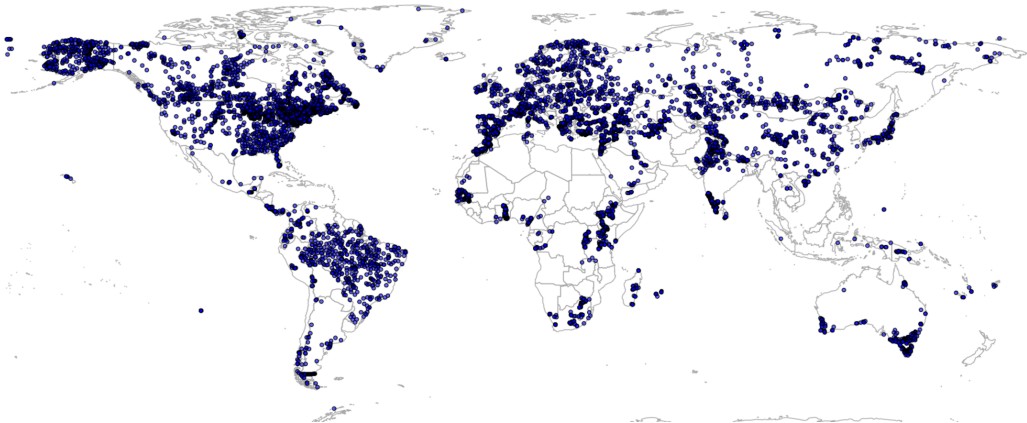

**Figure 1 Global spatial distribution of the BIOME 6000 dataset enriched by *Hengl et al. (2018)*.** Despite the added pseudo points, there are still large areas (Patagonia, Sahara region, Central Africa and most parts of Australia) not covered by any observation.

climate policies and other similar factors would evolve by 2100. A subset of the new 50 CMIP6 models has been considered overly sensitive (*i.e.,* "too hot") and with climate warming in response to carbon dioxide emissions that might be larger than supported by other evidence (*Hausfather et al., 2022*; *Zelinka et al., 2020*). For this reason, we decided to exclude CMIP6 models from our analysis and rely instead on CHELSA v.1.2 data with CMIP5 calculations, using an ensemble of 5 Global Circulation Models (GCMs): the Max-Planck-Institute Earth System Model (MPI-ESM-mr) (*Giorgetta et al., 2013*), the version 5 of the Model for Interdisciplinary Research on Climate (MIROC5) (*Watanabe et al., 2010*), the Community Earth System Model version 1 that includes the Community Atmospheric Model version 5 (CESM1-CAM5) (*Neale et al., 2010*), version 5 of the Institut Pierre Simon Laplace Coupled Model (IPSL-CM5A-MR) (*Dufresne et al., 2013*) and the First Institute of Oceanography-Earth System Model (FIO-ESM) (*Qiao et al., 2013*). Since most of the GCMs are interdependent between each other, and not all of them include the three RCP scenarios we analyzed in this study, we followed the suggestions of *Sanderson, Knutti & Caldwell (2015)* for the selection process.

To train the model, we used average values for the period 1979–2013 for 17 bioclimatic variables, *i.e.,* annual mean temperature, mean diurnal range, isothermality, temperature seasonality, maximum temperature of the warmest month, minimum temperature of the coldest month, temperature annual range, mean temperature of the wettest quarter, mean temperature of the driest quarter, mean temperature of the warmest quarter, mean temperature of the coldest quarter, annual precipitation, precipitation of the wettest month, precipitation of the driest month, precipitation of the wettest quarter, precipitation of the driest quarter, precipitation of the warmest quarter and precipitation of the coldest quarter. The precipitation seasonality (BIO15) was not included because of its excessive number of missing values in the layers of the future time periods. We also used monthly minimum, average and maximum temperature and monthly precipitation, for a total of 66 climatic and bioclimatic predictor variables. They can be downloaded

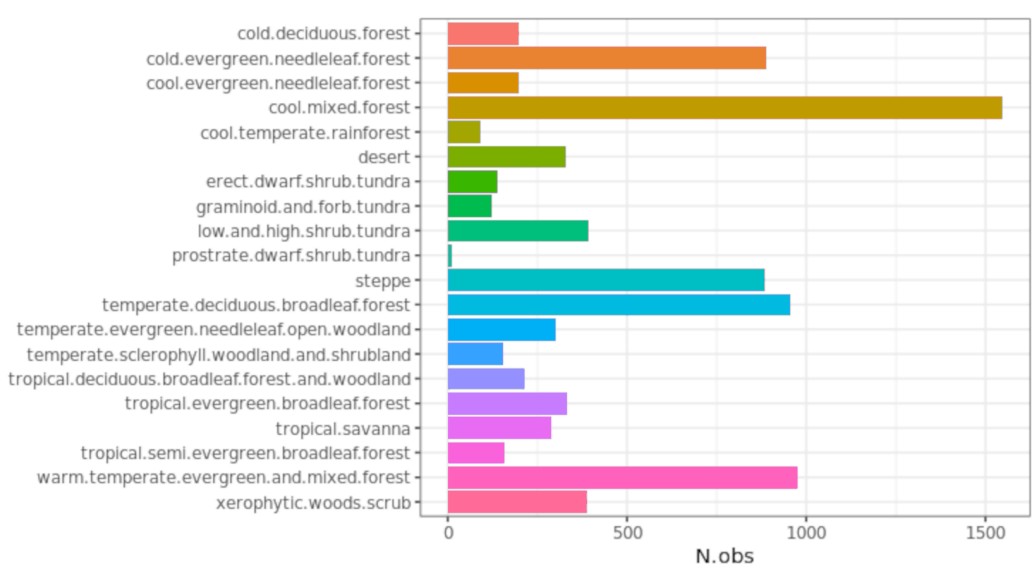

**Figure 2  Number of observations per biome class.** Note the strong imbalance between the different classes, with the most abundant class ("*cool mixed forest*") counting > ~1500 observations while the least abundant ("*prostrate dwarf shrub tundra*") counts < 20 observations.

from https://chelsa-climate.org/downloads/. We used six additional predictor variables representing topographic conditions, *i.e.,* sine and cosine of aspect, slope, upslope curvature and downslope curvature. These covariates have a three arcsecond resolution (~ 90 m at the equator) and they were derived from MERIT DEM (*Yamazaki et al., 2017*). MERIT DEM layers can be downloaded from http://hydro.iis.u-tokyo.ac.jp/~yamadai/MERIT_DEM/.

## Model building and evaluation

We used an ensemble machine learning approach based on stacked generalization (*Wolpert, 1992*). Ensemble modeling techniques involve training several independent models with the same input data and then aggregating each of the model outputs into the final predictions. Stacked generalization uses the outputs of the individual models to train an additional model (*meta-learner* from here on) which then produces the final predictions. We used Random Forests (RF) (*Breiman, 2001*), generalized linear models (*Nelder & Wedderburn, 1972*) with Lasso regularization (*Tibshirani, 1996*) and gradient-boosted trees (GBT) (*Friedman, 2002*) as component models for the ensemble model. To reduce overfitting in the training phase, we used a fivefold spatial cross validation (*Roberts et al., 2017*): the out-of-fold predictions were used to train the *meta-learner*. Spatial cross validation was implemented by a 100 × 100 km grid and using the tile ID as the blocking variable during the training of the models. We used multinomial logistic regression (*Wright, 1995*) as the *meta-learner*.

Predictions are delivered as probability maps (0–100%) together with uncertainty maps: the standard deviation of the predicted values by the base learners serves as an indication of model uncertainty. The principle is that the higher the standard deviation, the more uncertain the model is regarding the probability to be assigned to the pixel (*Brown,*

*Bhuiyan & Talbert, 2020*). In contrast, for the hard class map we used the probability maps to calculate a per-pixel confidence metric. Contrary to *Hengl et al. (2018)*, we chose not to use the per-pixel entropy (*Shannon, 1948*) but the margin of victory (*Calderón-Loor, Hadjikakou & Bryan, 2021*) The margin of victory is defined as the difference between the first and the second highest class probability value in a given pixel. Potential values in this case would go from 0 (*i.e.* no difference between the first two classes, highest confusion possible) to 100 (the model is certain in the class probability value attribution, no confusion with other classes); in short, high values would be measures of low uncertainty, while low values would indicate a high uncertainty. All the analysis were performed using R (version 4.1.1) (*R Core Team, 2021*) and, specifically, the *mlr* package (*Bischl et al., 2016*). For more details on the hyperparameter space used for the other component models and the overall architecture of the ensemble model, see *Bonannella et al. (2022)*.

We calculated the variable importance for each of the component models using Gini importance for RF, the gain metric for GBT (*Shi et al., 2019*) and the coefficients for the minimum value of $\lambda$ for GLM (*Hastie, Qian & Tay, 2016*): we took the 20 most important variables across the component models and retained the variables that these learners had in common. We then report these as the most important variables for the ensemble model. The predictive performance of the ensemble model was assessed through fivefold spatial cross validation repeated five times with overall accuracy and the $R^2_{\text{logloss}}$ (*Bonannella et al., 2022*) as performance metrics. We then computed the $R^2_{\text{logloss}}$ in addition to more classic metrics used for classification problems, like the True Positive Rate (TPR) and the F1 score (*Van Rijsbergen, 1979*) to assess model performances per class.

## Shifts in potential biomes

While for data-model comparison we used the original 20 classes classification system from *Hengl et al. (2018)*, to compare the model outputs we translated the classes in the IUCN Global Ecosystem Typology (*Keith et al., 2020*). This system classifies biomes based on functional characteristics and their structural role in the ecosystems rather than on climate, species distribution or vegetation patterns. Its principle is very similar to that of the BIOME 6000 classification system *Prentice & Webb III (1998)*. The IUCN system comprises six hierarchical levels, with the three upper ones being *realms*, *biomes* and *functional groups*: the definitions of the functional groups are quite different from those of BIOME 6000, so we aggregated the 20 classes used in this study at the *biome* level according to the IUCN. We focused on the biomes present in the *terrestrial* realm, which include the following:

- **T1** - Tropical-subtropical forests biome;
- **T2** - Tempereate-boreal forests and woodlands biome;
- **T3** - Shrublands and shrubby woodlands biome;
- **T4** - Savannas and grasslands biome;
- **T5** - Deserts and semi-deserts biome;
- **T6** - Polar/alpine (cryogenic) biome;
- **T7** - Intensive land-use biome.

**Table 1** Overview of the translation scheme used to pass from BIOME 6000 to IUCN classes.

| BIOME 6000 class (from *Hengl et al. (2018)*) | IUCN class |
|---|---|
| Tropical deciduous broadleaf forest and woodland | T1 - Tropical-subtropical forest biome |
| Tropical evergreen broadleaf forest | |
| Tropical semi evergreen broadleaf forest | |
| Cold deciduous forest | T2 - Temperate-boreal forests and woodlands biome |
| Cold evergreen needleleaf forest | |
| Cool evergreen needleleaf forest | |
| Cool mixed forest | |
| Cool temperate rainforest | |
| Temperate deciduous broadleaf forest | |
| Temperate sclerophyll woodland and shrubland | |
| Temperate evergreen needleleaf open woodland | T3 - Shrublands and shrubby woodland biome |
| Warm temperate evergreen and mixed forest | |
| Xerophytic woods scrub | |
| Tropical savanna | T4 - Savannas and grassland biome |
| Desert | T5 - Desert and semi-desert biomes |
| Steppe | |
| Erect dwarf shrub tundra | T6 - Polar/alpine (cryogenic) biome |
| Graminoid and forb tundra | |
| Low and high shrub tundra | |
| Prostrate dwarf shurb tundra | |

Since the focus of this paper is on potential biomes, the "*T7 - Intensive land-use biome*" class was not considered. The complete translation scheme is available in Table 1: we calculated the IUCN class by aggregating the per-class probability values of the BIOME 6000 classes according to the translation scheme. We computed the margin of victory for the IUCN classes as well and we used those maps to highlight areas with high confidence (*i.e.,* low confusion) predictions. To assess change in potential biome class in fact, we calculated the difference in hard class between the potential biomes map of the current period and each of the future periods and RCP scenarios. We first reprojected all the IUCN classes and relative margin of victory maps to the Interrupted Goode Homolosine projection, which is an equal-area composite projection. We chose it specifically to provide an unbiased (*i.e.,* without geographical distortions) estimate of the areas subjected to change. In the results we discuss change dynamics only for the aggregated IUCN classes and only for pixels having a margin of victory $\geq 50\%$; pixels with a margin of victory $< 50\%$ are not considered.

## RESULTS

### Model performances and variable importance

The hyperparameter tuning resulted in the following architecture for the ensemble model:

- Random Forest: 452 trees, minimum node size 9, *mtry* 10, while the other hyperparameters were set to default;

**Table 2  Results of the repeated fivefold spatial cross validation per class.**

| Class | N.obs | TPR | F1 | $R^2_{logloss}$ |
|---|---|---|---|---|
| Cold deciduous forest | 199 | 0.51 | 0.57 | 0.53 |
| Cold evergreen needleleaf forest | 890 | 0.78 | 0.76 | 0.62 |
| Cool evergreen needleleaf forest | 198 | 0.23 | 0.31 | 0.30 |
| Cool mixed forest | 1548 | 0.81 | 0.79 | 0.62 |
| Cool temperate rainforest | 93 | 0.66 | 0.70 | 0.59 |
| Desert | 328 | 0.51 | 0.55 | 0.50 |
| Erect dwarf shrub tundra | 138 | 0.36 | 0.42 | 0.50 |
| Graminoid and forb tundra | 123 | 0.41 | 0.49 | 0.36 |
| Low and high shrub tundra | 391 | 0.68 | 0.66 | 0.63 |
| Prostrate dwarf shrub tundra | 11 | 0.00 | 0.00 | −0.09 |
| Steppe | 884 | 0.67 | 0.66 | 0.46 |
| Temperate deciduous broadleaf forest | 958 | 0.62 | 0.62 | 0.47 |
| Temperate evergreen needleleaf open woodland | 302 | 0.58 | 0.59 | 0.52 |
| Temperate sclerophyll woodland and shrubland | 153 | 0.76 | 0.74 | 0.71 |
| Tropical deciduous broadleaf forest and woodland | 215 | 0.42 | 0.47 | 0.49 |
| Tropical evergreen broadleaf forest | 333 | 0.79 | 0.77 | 0.74 |
| Tropical savanna | 291 | 0.77 | 0.71 | 0.67 |
| Tropical semi evergreen broadleaf forest | 160 | 0.40 | 0.43 | 0.54 |
| Warm temperate evergreen and mixed forest | 976 | 0.73 | 0.67 | 0.52 |
| Xerophytic woods scrub | 387 | 0.45 | 0.48 | 0.42 |

- Gradient boosted trees: 20 boosting rounds, maximum depth per tree 5, learning rate 0.5, minimum loss reduction to split a leaf node 10, subsample ratio of the training instances 1, subsample ratio of columns when constructing each tree 0.5. The other hyperparameters were set to their defaults;
- Generalized Linear Models with Lasso: λ value $1.1 \times 10^{-5}$;
- Multinomial logistic regression: multinomial function to minimize the loss.

The ensemble model had a moderate accuracy; according to the fivefold spatial cross validation the overall accuracy is 0.67 and the $R^2_{logloss}$ 0.61. Model performances per class are shown in Table 2. The "*tropical evergreen broadleaf forest*" is the class with the greatest gain in predictive performances ($R^2_{logloss} = 0.74$) compared to the baseline logloss, while the "*prostrate dwarf shrub tundra*" is the worst predicted class, with a negative gain in predictive performances compared to the baseline logloss (see Fig. 3).

The latter may be attributed to the very small (n.obs = 11) number of points in the training data for this specific class. It is also the only class with negative gain in predictive performances: all the other classes go from weak ("*cool evergreen needleleaf forest*", $R^2_{logloss} = 0.30$) to consistent ("*temperate sclerophyll woodland and shrubland*", $R^2_{logloss} = 0.71$) increase in predictive performances. The three models captured different parts of the feature space despite the relatively few (72) number of predictor variables. From the top-20 predictor variables, only one is shared across all component models, BIO2, the mean diurnal range. RF was the only component model which selected a

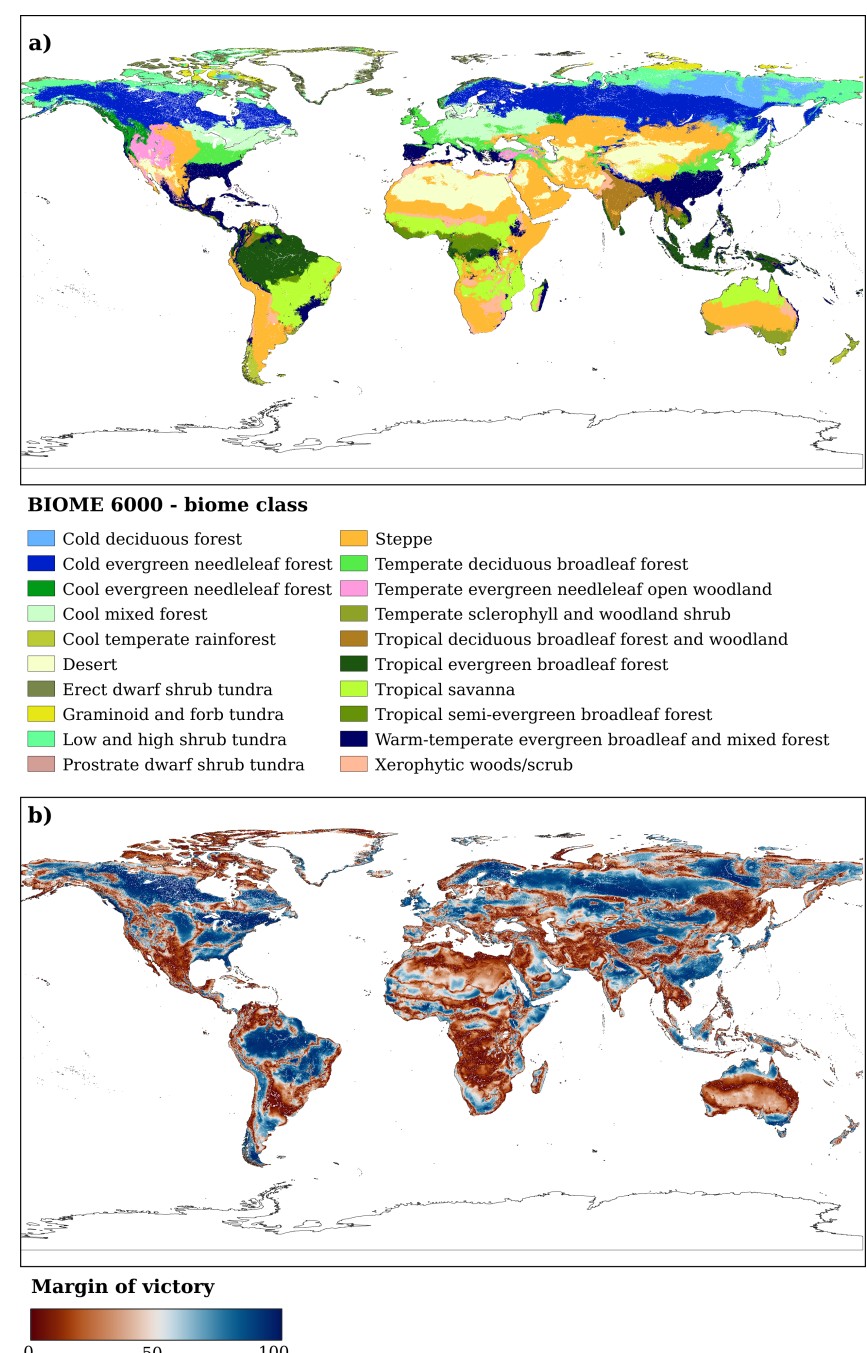

**Figure 3** Distribution of (A) the biome classes according to the BIOME 6000 classification scheme and (B) the margin of victory for the current time period. The margin of victory is here used as an indication of uncertainty. High values (blue in figure) indicate high confidence in the attribution of dominant class by the model, while low values indicate high uncertainty.

topographic predictor (elevation) as one of the most important variables, while the other two models focused mostly on the climatic variables. RF and GBT shared nine out of 20 predictor variables, with seven out of nine being temperature-related (mean or maximum temperature and temperature-derived bioclimatic variables). GLM with Lasso differed mostly from the other two component models in the selected most important predictor variables. GLM was the only component model selecting variables from the group of the minimum temperatures.

## Future predictions

Examining the biome state transitions from the current conditions to the future epochs, we found that most locations remained stable. Filtering the transitional areas with the margin of victory across all scenarios and epochs using 50% as a safety threshold value considerably reduced the predicted transitional area: less than 1% of the Earth's surface showed signs of change, with the least changes found in the scenario RCP 2.6 ($5.6 \times 10^5$ km$^2$) and the most in the scenario RCP 8.5 for the epoch 2061–2080 ($5.0 \times 10^6$ km$^2$). For epoch 2040–2060, the main changes shown by all three scenarios are as follows: areas belonging to the polar/alpine biome will transition to the temperate-boreal forest biome and areas from the tropical forest biome will transition to more drier biomes, like the savannas and grasslands biome the shrublands biome and, in some cases, the deserts and steppes biome.

The same tendency can be observed for the temperate-boreal forest biome, with the difference that transitional areas are almost equally split between the shrublands biome and the deserts and steppes biome. It is interesting to notice that almost all the transitioned pixels from the tropical forest biome would change to savannas and grasslands biome in the RCP 4.5 and RCP 8.5 scenarios, while for scenario RCP 2.6 one third would change to the shrublands biome (see Fig. 4). According to scenario RCP 2.6, most of the changes would happen in the polar/alpine biome, so at higher latitudes, while for the other two scenarios the tropical areas seem to be the ones most affected. For scenario RCP 2.6, the transitional areas are also more equally split across the different classes, while for scenarios RCP 4.5 and RCP 8.5 almost 50% and 60% of the transitional areas would shift to the savannas and grasslands biome.

Figure 5 shows the geographic locations of the biome shifts according to the three climatic scenarios. It is possible to discern different clusters where the changes are located: the most noticeable is in the tropical area, between the Equator and 15°S; in South America, the region affected corresponds with the southern edges of the Amazon rainforest, which would shift from a tropical forest biome to savanna. In Central Africa, in the contiguous borders of Angola, Congo and Zambia, the shift goes instead from shrubland or steppic biomes to savanna. In the transitioning areas where all the scenarios agree in predicting change, there is one area that includes most of the shifts: between 60° and 75°N, just around the Arctic Circle. At this latitude, most of the areas currently in the polar/alpine class would shift to the boreal/temperate forest class. Big clusters can be observed in the northern parts of Canada and Alaska, while smaller clusters occur in Scandinavia, European Russia and some areas in Siberia. In most cases, scenario RCP 2.6 involves the smallest amount of

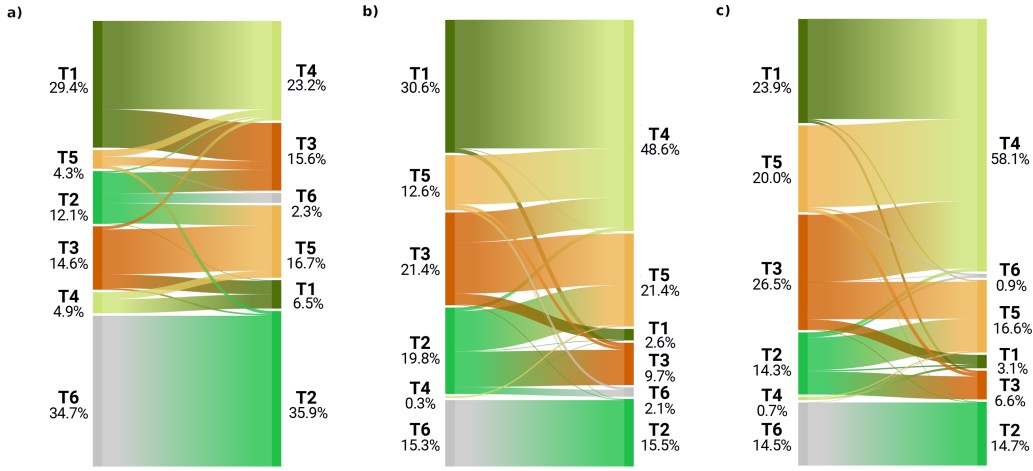

**Figure 4** **Biome transitions predicted for epoch 2040–2060 for the three climatic scenarios.** Only the pixels that transitioned are represented in this diagram, so the percentages represent different amounts of surface area across the scenarios. For each plot, on the left axis the proportion of transitioned pixels in the current conditions and on the right axis the final state according to each climatic scenario: (A) the transitional areas for scenario RCP 2.6, (B) for scenario RCP 4.5 and (C) for scenario RCP 8.5.

transitions, while RCP 8.5 involves the greatest. However, there are also some areas where this does not hold, as can be seen in Figs. 5C and 5F. In the first case, scenario RCP 2.6 involves the greatest amount, while in the latter it is greater than scenario RCP 4.5 but smaller than scenario 8.5. In general, all three scenarios agree in predicting a change in some 3% of the cases, while if we consider the two most radical scenarios the percentage rises to 11%.

For epoch 2061–2080, we found similar trends to the ones observed in the previous epoch: all of the pixels from the polar/alpine biome tend to shift to the temperate-boreal forest biome and the pixels from the tropical forest biome would shift towards the savannas and grassland biome (see Fig. 6). The tendency shown in Fig. 4A, with the transitioning pixels from the tropical forest biome split between the shrublands biome and the savannas and grassland biome, is in this epoch even more pronounced: the ratio is reversed, with one third of the pixels transitioning to the savannas and grassland biome and the rest towards the shrublands biome. In the other two scenarios, once again, almost all transitioning pixels from the tropical forest biome would shift to the savannas and grassland biome. Another recurring pattern is how most of the transitioning pixels from the temperate-boreal forest biome are equally split between the shrublands biome and the deserts and steppes biome, so either the canopy of those forests would become more open and the ratio between trees and shrub would increase in favor of the latter, or they become so dry that the trees are replaced by steppic vegetation. For scenario RCP 2.6, 80% of the transitioning pixels are part of the tropical forest biome, the temperate-boreal forest biome or the polar/alpine biome, with nearly 39% coming from the polar/alpine biome, while the classes which would gain most of this surface area are the temperate-boreal forest biome and the shrublands biome. In the other two scenarios, the polar/alpine biome covers a more marginal importance across
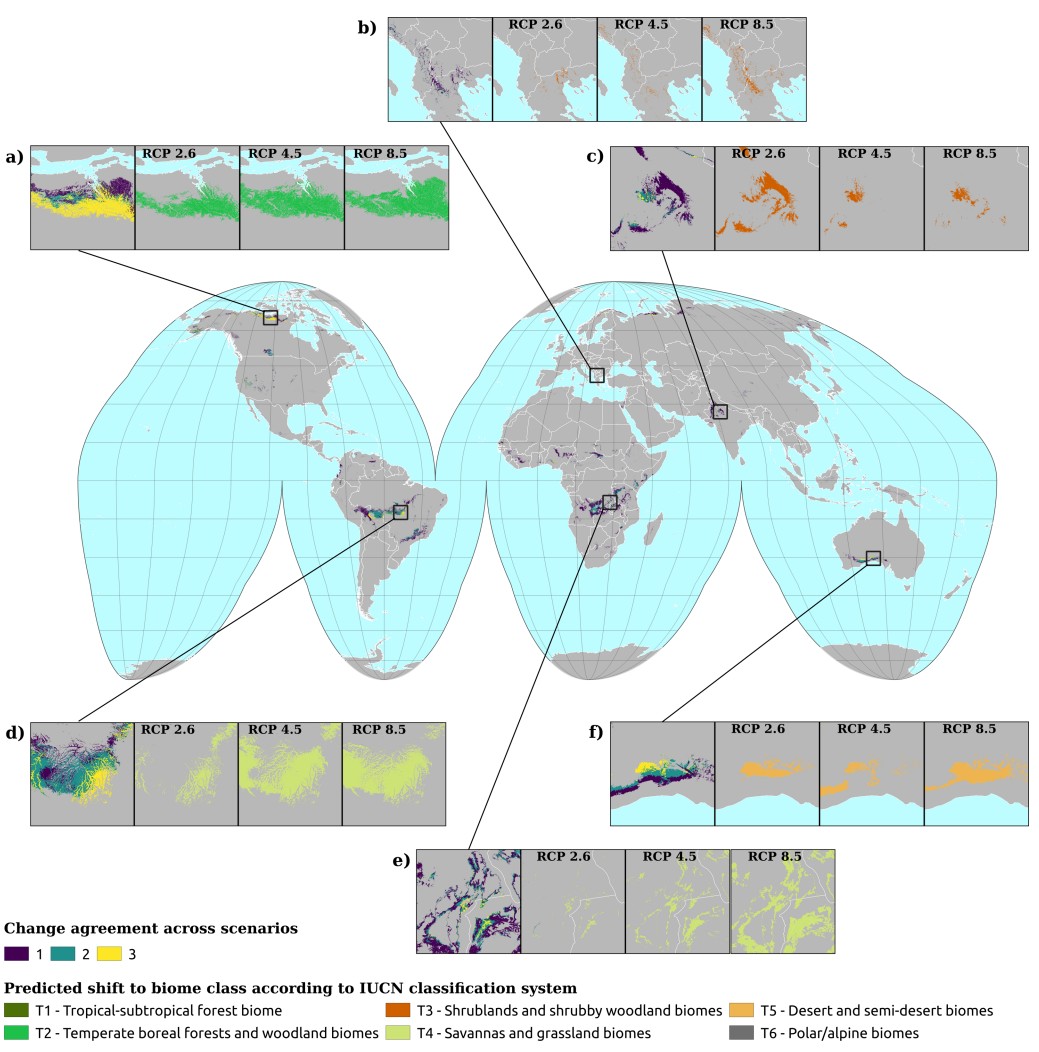

**Change agreement across scenarios**
■ 1 ■ 2 ■ 3

**Predicted shift to biome class according to IUCN classification system**
■ T1 - Tropical-subtropical forest biome
■ T2 - Temperate boreal forests and woodland biomes
■ T3 - Shrublands and shrubby woodland biomes
■ T4 - Savannas and grassland biomes
■ T5 - Desert and semi-desert biomes
■ T6 - Polar/alpine biomes

**Figure 5** **Spatial location of biome transitions as predicted by our ensemble model according to the three climatic scenarios for epoch 2040–2060.** Colors on the main map show the degree of agreement between the three climatic scenarios: a value of 1 means that only one of the scenarios considers the pixel as transitioning, while a value of 3 shows complete agreement across the three scenarios. Inserts show towards which biome the current pixels are transitioning to according to the different scenarios. Inserts (A, D and E) show the main trends, with transitions from T6 (polar) to T2 (boreal forest) in (A) and from T1 (tropical forest) to T4 (savannas) in (D and E).

the transitioning pixels: it is the third most important. Some differences can be observed in the transitioned classes as well: in scenario RCP 2.6, about 40% of the pixels go to the temperate-boreal forest biome, while in the other two scenarios the savannas and grassland biome takes from 55% to 60% of the transitioned classes.

Figure 7 shows the geographic locations of the shifts for the epoch 2061–2080. The two big clusters observed in the previous epoch remain, as well as the area around the Arctic Circle; the transitioning pixels where all three scenarios agree in predicting change are also located mostly in these two areas. Compared to the previous epoch, more small clusters of
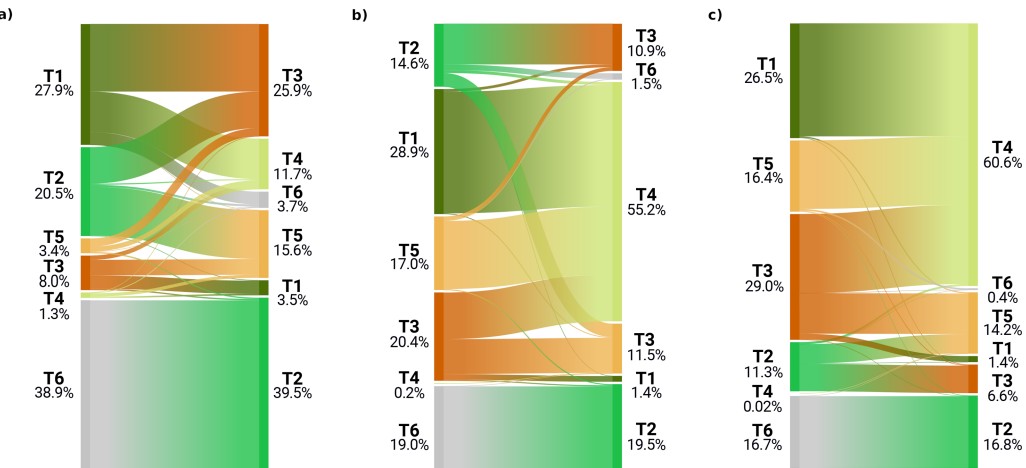

**Figure 6** **Biome transitions predicted for epoch 2061–2080 for the three climatic scenarios.** Only the pixels that transitioned are represented in this diagram, so the percentages represent different amounts of surface area across the scenarios. For each plot, on the left axis the proportion of transitioned pixels in the current conditions and on the right axis the final state according to each climatic scenario: (A) the transitional areas for scenario RCP 2.6, (B) for scenario RCP 4.5 and (C) for scenario RCP 8.5.

pixels, with no specific spatial pattern, are visible in the African continent from the Equator to 15°N (mostly around the Gulf of Guinea), Western India and Mediterranean Europe (mostly around the Pyrenees, see Fig. 7B); by checking the probability layers, it is possible to see the gradual shift (see Fig. 8) in vegetation conditions over time.

Contrary to the previous epoch, the mapped changes in the scenarios seem to be more consistent: all the inserts in Fig. 7 show that scenario RCP 2.6 is the one which projects the smallest number of transitioning pixels, while RCP 8.5 projects the most. The scenarios in this epoch also have a higher degree of agreement: all three scenarios agree in considering as changing 7% of all the transitioning pixels, while the value for agreement between two scenarios is just 8%. In general, if in the previous epoch most of the transitioning pixels were located either in the tropics or around the Arctic Circle, in this epoch we see them appearing in the temperate areas as well.

# DISCUSSION

## Model evaluation and comparison with previous works

In this study we trained an ensemble machine learning model to classify the current and future potential distribution of biomes under different climate change scenarios. Our results show that it is possible to produce relatively accurate maps of natural vegetation using ensemble machine learning approaches and to reach consistent accuracy values even with a limited selection of predictor variables. Comparing our results with the previous work of *Hengl et al. (2018)*, we achieved an increase in the overall accuracy by using an ensemble model and only 72 instead of 158 predictor variables: in the previous task the most accurate model (Random Forest) shows a spatial cross–validation overall accuracy of 0.33, less than half of what we estimate with the improved model (0.67). The performance

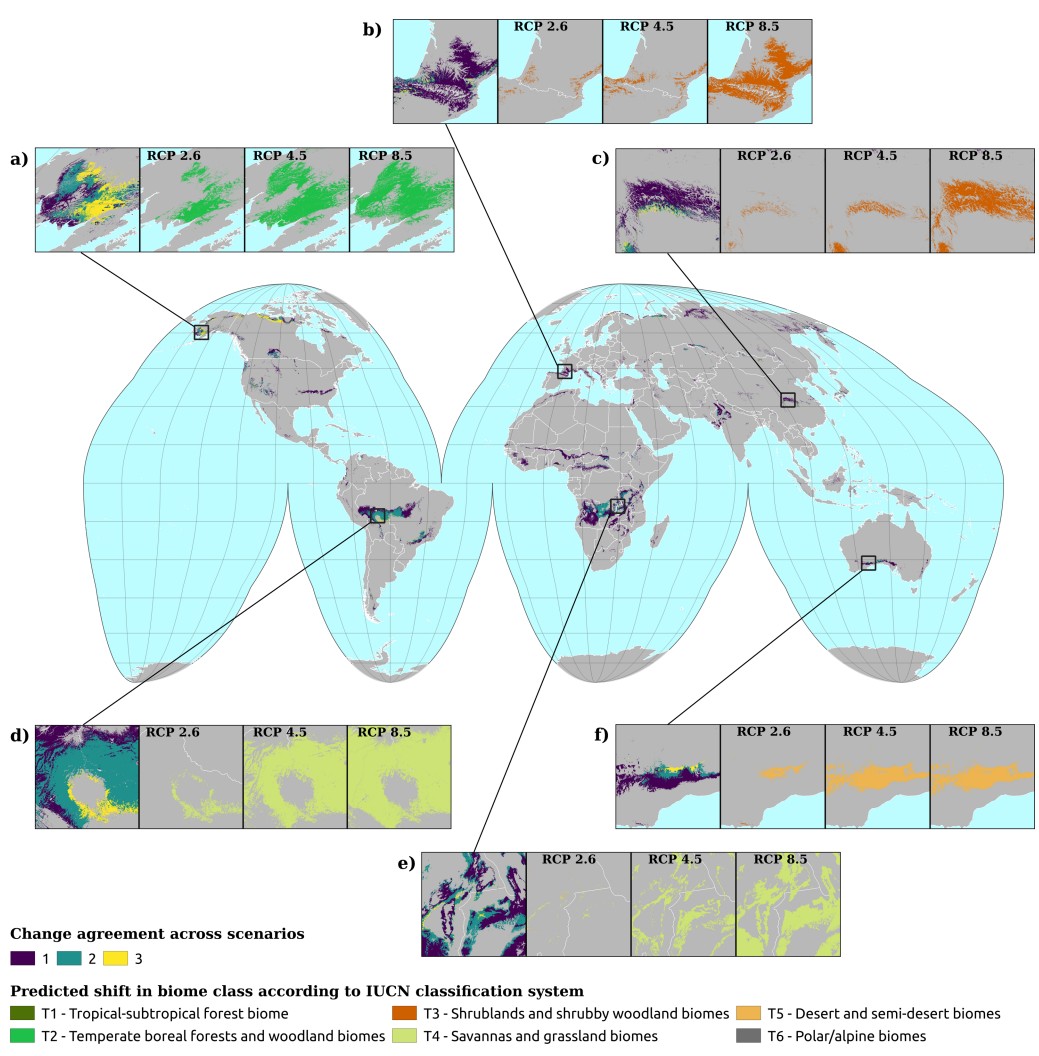

**Figure 7** **Spatial location of biome transitions as projected by our ensemble model according to the three climatic scenarios for epoch 2061–2080.** Colors on the main map show the degree of agreement between the three climatic scenarios: a value of 1 means that only one of the scenarios considers the pixel as transitioning, while a value of 3 shows complete agreement across the three scenarios. Inserts show towards which biome the current pixels are transitioning according to the different scenarios. Inserts (A, D and E) show the main trends, with transitions from T6 (polar) to T2 (boreal forest) in (A) and from T1 (tropical forest) to T4 (savannas) in (D and E) Inserts (B and C) show instead the tendency to drier ecosystems in temperate areas.

values per class show some degree of agreement: when comparing TPR values both studies consider the *"prostrate dwarf shrub tundra"* class as the worst predicted, while they disagree in the best predicted class (*"temperate sclerophyll woodland and shrubland"* for *Hengl et al. (2018)*, *"cool mixed forest"* in this study); while our TPR values are consistently lower across all classes, TPR values reported by *Hengl et al. (2018)* are for the model with no spatial partitioning. Model outputs are provided in probability values and not as hard classes in both studies; furthermore, the dataset is heavily imbalanced, as shown in Fig. 2. This makes logloss, and the $R^2_{logloss}$, a better metric to report model performances since it

**2022–2023 (current)**    **2040–2060 (RCP 4.5)**    **2061–2080 (RCP 4.5)**

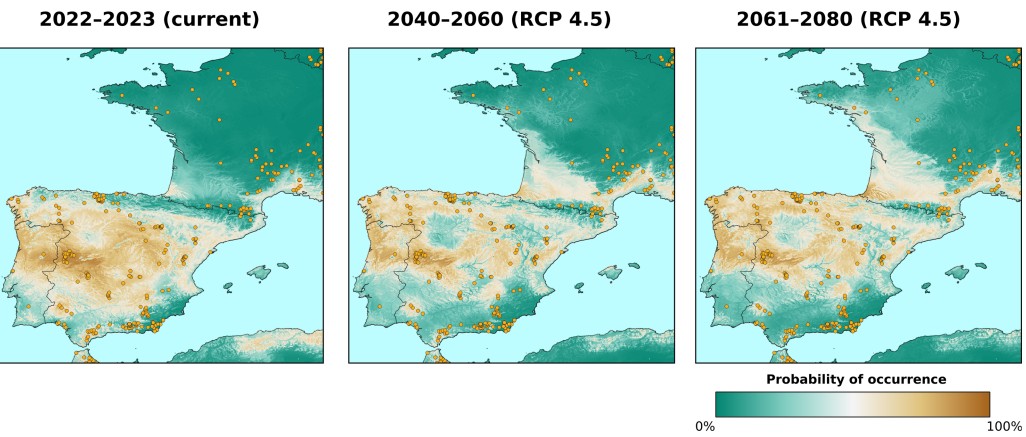

**Probability of occurrence**

0%                   100%

**Figure 8  Predicted probability of occurrence of *"warm temperate evergreen and mixed forest"* class, zoom in on the area around the Pyrenees.**  The probability values over time show that the class is slowly shifting towards northern latitudes. Only the RCP 4.5 scenario is shown since it is considered as the "middle of the road" scenario. Points indicate training points from the BIOME 6000 dataset.

indicates how close the predicted value for an observation and its respective label is; logloss is also one of the most robust performance metrics when it comes to imbalanced data (*Ferri, Hernández-Orallo & Modroiu, 2009*). The fact that *Hengl et al. (2018)* used other performance metrics that do not fit the task at hand to report per class results, may have caused an overestimation of those values.

In machine learning, increasing the size of the feature space is expected to provide more discriminating power (*Hall & Holmes, 2003*), at the cost of higher computation time. However, it can also increase the complexity of the task at hand to the point that the added information is redundant or introduces noise in the model (*Bellman & Kalaba, 1957*). While feature selection procedures help considerably in tackling this problem, in this case we used expert knowledge to select only climatic and topographic predictor variables. By doing that, we managed to achieve a twofold goal: reduce task complexity (*i.e.,* less features) while maintaining consistent values of accuracy, and keeping the model simple enough to be able to transfer it to future epochs without introducing too many assumptions in the modeling framework; a similar approach to calculate future projections was used by *Anjos et al. (2021)*, *Maksic et al. (2022)* and *Zevallos & Lavado-Casimiro (2022)*, respectively, for the whole South America, Brazil and Peru. In the case of *Zevallos & Lavado-Casimiro (2022)*, they used a Random Forest model and achieved higher levels of accuracy; however, they trained it on a smaller (six bioclimatic) set of predictor variables, used a 80:20 train test split and did not use any spatial partitioning. Considering the huge differences in accuracy in the results obtained by *Hengl et al. (2018)* in their Random Forest model with and without spatial partitioning, there is the risk that predictions from *Zevallos & Lavado-Casimiro (2022)* may have been optimistic; on the other hand, their analysis is focused on just one country and not on a global scale, so it is still possible to reach high levels of accuracy on a limited study area.

Data-driven approaches in this topic mostly deploy Random Forest models, with predictions with high levels of agreement with process-based models: *Lindgren et al. (2021)* used Random Forest to reconstruct past global vegetation and compared their results with the LPJ-GUESS global dynamic vegetation model; the model was able to produce comparable results to the LPJ-GUESS when enough training data was available, with bad performances when predicting in the Last Glacial Maximum, the epoch with the least training data. Extrapolation and transferability are two common issues of machine learning models and data-driven approaches in general, which have limited the reliability of such methods in environmental modeling(*i.e.,* invasive species modeling, past vegetation reconstruction, future vegetation forecasting etc.) (*Qiao et al., 2019*). However, ensemble modeling is known to provide more advantages than using individual machine learning models, since ensemble models reduce model uncertainty (*Bonannella et al., 2022*; *Mehra et al., 2019*). For future projections, "ensemble datasets" are more common than ensemble models: climate is assumed to be the major driving force for large-scale vegetation patterns (*Whittaker & Marks, 1975*); starting from this assumption, multiple studies create the training dataset by averaging together temperature and precipitation values as calculated by different GCM simulations (*Beigaite et al., 2022*; *Anjos et al., 2021*), hence why we chose an ensemble of five independent GCMs. The model used in this study could benefit from using such datasets: while studies on performance comparisons between the different GCM simulations are available for the CMIP5 project (*Sanderson, Knutti & Caldwell, 2015*), the same cannot be said for the new CMIP6 simulations; future applications of the experimental design presented in this study that would use CMIP6 simulations, could benefit from using an ensemble dataset of all 50 of the GCM simulations instead of relying on the data of only five models.

## Biome shifts: key emerging trends

We evaluated changes in potential biomes in two future epochs and across three different climatic scenarios: our results show that the distribution of the biomes on land in the future will mostly ($\geq$99% land surface) remain the same. The limited geographic extent of the biome shifts under all three scenarios has probably to do with the chosen conservative threshold in margin of victory; despite that, the projections show specific emerging trends in biome shifts in precise locations of the globe that, while differing in size, are common across all the climatic scenarios.

One of these emerging trends is the transition from a polar to a boreal forest biome in the global north, around the Arctic Circle: in both of the epochs analyzed in the study, it is one of the most evident and consistent transitions in all three climatic scenarios, with its extension increasing in epoch 2061–2080. According to the future climatic projections, all three scenarios forecast either a modest or consistent rise in temperatures by 2100, from well below 2°C for scenario 2.6 and around 5°C for scenario RCP 8.5. Areas where this change was predicted currently present vegetation not belonging to the "*boreal forests and woodland biomes*" class mostly due to the fact that the low temperatures are a limiting factor for the presence of trees. The hypothesis that the thawing of permafrost would lead to the tree line advancing towards the North Pole finds more and more evidence,

the last one provided by *Berner & Goetz (2022)* using Landsat time-series: their results on vegetation greenness for the period 1985–2019 showed a prevalence of greening in the pan-boreal vegetation. It is important to point out how this phenomenon is not uniform across the circumpolar arctic vegetation: while greening was more prevalent than browning phenomena, browning was still predicted in those areas where summers have become warmer and drier in the last 40 years. Our results, regardless of the scenario, show that while the change will happen, it will not be uniform across North America, Asia and Europe: North America seems to be the one that will be most affected, while only in few areas of Siberia all scenarios agree in a biome shift. Our maps can be used to locate *hot spots* of change, from which the shift can then expand: the model used in this study does not take into account many factors, such as the feedback effects on the carbon cycle caused by the permafrost thaw (*Smith et al., 2022*) or soil temperature, moisture and content, which, in turn, affect vegetation productivity and functional types (*Berner et al., 2020*). For this reason it is important to be cautious when assessing biome change implications. On top of that, we focus on the potential conditions that define a biome and not on what is currently on the ground: species that live in a biome may not be able to keep pace with the climate change advance (*Rees et al., 2020*), and while our maps may show that the conditions for a shift are present in a specific location, its reality may be different.

The second trend relates to the transition from tropical forest to savannas and grasslands, in particular in the southern edges of the Amazon and the Congo rainforests. Both the entity and the extension of this shift across the climatic scenarios follow the same pattern that we found for the transition from polar to boreal, with the highest value of pixels shifting found in the epoch 2061–2080 for scenario RCP 8.5. The consequences of climate change on the Amazon rainforest are a critical area of research given its importance for global climate regulation and biodiversity (*Foley et al., 2007*; *Lawton, 1998*) and have been the subject of extensive research in the scientific community. While the full extent of these changes has not yet been completely understood, higher temperatures and variations in rainfall regime have been causing longer and more severe dry seasons (*Xu et al., 2022*; *Agudelo et al., 2019*; *Arias et al., 2015*), with an increase in frequency of droughts, floods and fires (*Barlow et al., 2020*; *Lovejoy & Nobre, 2018*; *Marengo & Espinoza, 2016*); a recent study by *Gatti et al. (2021)* has demonstrated how the southeastern edge of the Amazon rainforest has already reached the tipping point, acting as a net carbon source instead of a carbon sink. These findings agree with the projections showed in our results, which now are part of a long series of studies showing alarming signs of an incoming process of savannization in the area; the feedback loop created by a disruption in the carbon cycle such as the one showed by *Gatti et al. (2021)* could further exacerbate the savannization process. On the same note, *Sampaio et al. (2007)* were the first to show how when deforestation exceeds 40%, the savanna would become the new stable state of the ecosystem in south, east and partially central Amazonia due to altered precipitation patterns; climatological projections from *Higgins, Buitenwerf & Moncrieff (2016)* show how a rise by only 2°C in average temperature could lead to a loss of 50% of suitable areas for forest specialist species and an increase by 11%–30% for savanna species. While less studies on the Congo rainforest are available in literature, the projected savannization process can be attributed to the same causes:

*Giresse, Maley & Chepstow-Lusty (2023)* have shown that the Congo rainforest has been resistant to change in the last 1,000 years and it was not possible to identify any serious human impact over this period. However, the current increasing mix of climate change and human pressures (deforestation, agriculture expansion and other factors) may lead to unforeseen consequences: the current rainfall regime of much of the African rainforests is close to a threshold that favours savannas over rainforests (*Malhi et al., 2013*), so even a small alteration of this regime can cause large scale changes in the rainforest-savanna transition zone.

Overall, the shifts in biomes identified in this study portray a picture of minor or consistent changes across all the different biomes on the planet due to either an increase in temperature or decrease in precipitation/moisture conditions: this agrees with a similar existing dominating browning trend for global vegetation recently identified by *Higgins, Conradi & Muhoko (2023)* for the last four decades (1982–2015). Their approach is particularly relevant for the context of the present study, since they combined a process-based approach by using a dynamic plant growth model adjusted for climate with a data-driven approach by using Advanced Very High Resolution Radiometer (AVHRR) NDVI and EVI time series to describe vegetation activity. These shifts may have significant ecological implications for the distribution and diversity of plant and animal species, as well as societal implications for human communities that depend on these ecosystems. It is likely that these shifts will also have economic impacts, as the distribution of resources such as timber, livestock grazing and agriculture or the potential displacement of human population; the alterations of the services that these ecosystems provide, such as climate regulation and flood control, have also to be considered. These shifts may contribute to the loss of biodiversity, as some species may not be able to adapt to the new conditions in their range: the loss of tropical and subtropical biomes can have a negative impact on the species that depend on these biomes for their survival, as many species have narrow habitat requirements and are not able to adapt to changes in their environment. The expansion of boreal biomes, on the other hand, can have a positive impact on the species that depend on these biomes, as they will have access to new areas with suitable habitat.

### Technical limitations

There are some limitations to this study that should be considered. First, the dataset used to train the model was heavily imbalanced, with some classes having a very small number of observations: this has significantly affected the model performances for certain classes; on top of that, some locations are underrepresented, with limited or no observations. This is a serious limitation of the study, as the model may not be able to accurately predict the vegetation in those locations due to a lack of data. This highlights the importance of gathering more comprehensive ground truth data in the future to improve the model's accuracy and prediction abilities in those locations.

Secondly, while the use of expert knowledge to select the predictor variables let us to reduce the complexity of the task, it may also have introduced biases or limitations to the model's ability to accurately represent the full range of conditions present in different biomes; feedback loops (vegetation–climate interactions) or anthropogenic factors (human

disturbances, deforestation) were also not considered in the study. One of these effects which are difficult to incorporate in data-driven models is the fertilization effect of increasing concentration of $CO_2$ in the atmosphere, or $CO_2$ fertilization effect (CFE): process-based models are known to be able to include and parametrize this factor, event though the mechanisms and limits of it are still not completely understood. *Ballantyne et al. (2012)* observed that since the 1960s the carbon uptake by both terrestrial and oceanic ecosystems has increased instead of declining, while a study by *Chen et al. (2022)* calculated how approximately 44% of the increase in global Gross Primary Productivity (GPP) since the 2000s can be attributed to the CFE. This proved to be especially important for the tropical region, with different satellite-based or in-situ studies showing patterns of greening in these areas (*Anchang et al., 2019*; *Stevens et al., 2017*). Thus, not including this effect in our model may lead to overestimate the amount and the type of shifts in the tropical region: for example, the chapter on the African continent of the last IPCC report (*Trisos et al., 2022*) mentions an overall continental trend in woody plant expansion, especially in the non-arid areas, with high confidence that the trend is attributable to the CFE. This is in contrast with the desertification and contraction trend that was instead highlighted by the previous AR: since the AR6 had at its disposal longer time-series of observations, the trend captured by the previous AR, and hence the CMIP5 GCM projections, may have been overly pessimistic in some regions. Not including the CFE in our study and the usage of CMIP5 projections may have biased the results in favor of a desertification trend for some areas. All the simulations from the study from *Friend et al. (2014)* predicted a consistent increasing trend in $CO_2$ for every RCPs: so while our model may accurately capture a shift in the boreal region, due to the fact that the limiting factor for those biomes is temperature, it may capture a different type or extent of shifts in the tropical region, where neither temperature or precipitation are the limiting factors. On the other hand, however, *Higgins, Conradi & Muhoko (2023)* found the contribution to regreening from the CFE to be limited and *Wang et al. (2020)* observed a significant decline in the CFE on a global scale for the period 1982–2015. Overall, even if in this study we identified several *hot spots* of change in the tropical region, there is still high uncertainty for the future of biomes and their shifts there.

We also excluded from the study area those locations that are currently covered by permanent ice: glacial retreats due to climate change is a well known issue and the exclusion of these areas may also underestimate the potential changes in biomes in surrounding areas that may be influenced by the loss of glaciers. It is important to consider the inclusion of these areas in future studies to obtain a more comprehensive understanding of the potential impacts of climate change on the distribution of biomes. Another limitation is that the model cannot be considered spatiotemporal: while spatial relationships are taken into account during the modeling through the use of spatial partitioning, the temporal relationship is not considered since the model is trained on only one point in time. The model does not know that the three epochs analyzed in the study have an order in the temporal dimension: the phenomenon we want to predict at location $x$ for epoch 2061–2080 is not only a function of the predictor variables in the features space, but also of the realization of the phenomenon in previous states. The result, in the worst case scenario,

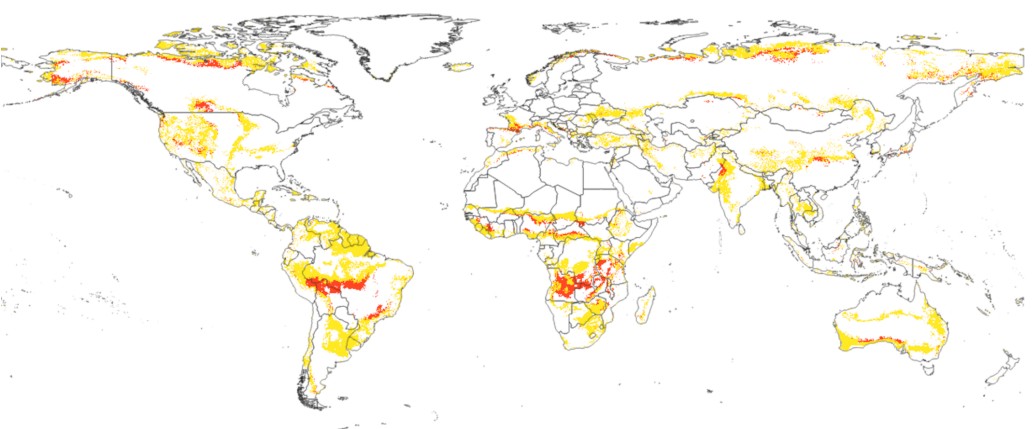

**Figure 9** **Spatial location of biome transitions for scenario RCP 8.** Five epoch 2061–2080. In red: all the areas that will change according to our model with a *margin of victory* value ≥50%. In yellow: all the transitioning areas with *margin of victory* values <50%.

is that the predicted values for one location may not be reliable over time: an example in our case would be a pixel labeled as *"desert"* for the current epoch, *"forest"* in the epoch 2040–2060 and *"desert"* once again in epoch 2061–2080, something that can be perfectly explained from a mathematical standpoint, but highly questionable from an ecological perspective.

Together with the heavily imbalanced dataset, this is another important reason to use the margin of victory to analyze the results. The inclusion of the margin of victory allows users to more accurately interpret the predicted maps and make more informed decisions based on the data. Without the margin of victory, the probability outputs could potentially be overinterpreted, leading to incorrect conclusions. The classification task examined in this study presented a total of 20 classes, with the model output per class constrained, for each pixel, to sum to 100%; the conservative threshold allowed us to focus our analysis only on those areas where the model found considerable differences in probability output between the dominant class and the remaining classes (see Fig. 9). Even though the class probabilities in this study are model-based predictions and model fit is far from perfect, the margin of victory gives an impression of the uncertainty in our projections. We acknowledge our inability to predict the future and to make claims about biome shifts that would certainly happen. It is however possible to indicate the confidence of our projections through the uncertainty layers provided. That's why we recommend to use the uncertainty layers to filter the predicted areas using conservative (≥50%) thresholds in case of future use of these maps in other works: false positives may thus be avoided while still identifying the main patterns.

## CONCLUSION

In this article we applied a methodological framework to predict current and future potential distribution of biomes under different climatic change scenarios using an ensemble machine learning approach. We focused our efforts on improving the caveats of previous

work from *Hengl et al. (2018)*, achieving greater accuracy in predicting current biomes and providing future distribution of biomes along with measures of prediction uncertainty to correctly interpret and use the maps. In general, our ensemble model achieved fairly accurate (overall accuracy = 0.67, $R^2_{\mathrm{logloss}} = 0.61$) results. Using expert knowledge to select only a limited number of predictor variables, we were able to achieve reasonable accuracy values while keeping the model simple enough to be able to transfer it to future epochs without introducing too many assumptions. Temperature-related predictor variables were considered as the most important to produce accurate predictions. Overall, this study demonstrates that an ensemble machine learning approach can be effective in modeling the potential distribution of biomes on a global scale and in identifying areas where climatological changes could lead to shifts in the distribution of these biomes.

Even though relatively small shifts in the distribution of biomes were projected under the RCP 2.6 and RCP 4.5 when compared to RCP 8.5, one of the significant findings of this study was the identification of areas where the change in climatological conditions could lead to a shift in the potential distribution of biomes regardless from which of the emission pathways analyzed will happen in the future. The biomes expected to shift the most are the tropical and subtropical biomes, particularly the tropical rainforests: these biomes are expected to experience a decrease in their potential distribution in the future time periods towards savanna and grassland biomes, a process called "*savannization*". In contrast, biomes located at higher latitudes, such as boreal forests, are expected to experience an expansion in their potential distribution in the future time periods at the expense of the polar biomes.

Further research is needed to better understand the factors that drive these shifts and the potential consequences for the distribution and diversity of plant and animal species, as well as for human communities. We hope that this study will contribute to the broader field of study by providing a framework that can be used to better understand the potential impacts of climate change on the distribution of biomes and their associated ecosystems, and by identifying areas where these impacts could be particularly significant. We recommend that this information is used by policy makers and land managers to make informed decisions about the management and conservation of these ecosystems, and to take action to mitigate the negative consequences of climate change.

### Funding
This work has been developed for the Open-Earth-Monitor Cyberinfrastructure project. The Open-Earth-Monitor Cyberinfrastructure project has received funding from the European Union's Horizon Europe research and innovation programme under grant agreement No. 101059548. The funders had no role in study design, data collection and analysis, decision to publish, or preparation of the manuscript.

### Grant Disclosures
The following grant information was disclosed by the authors:

The Open-Earth-Monitor Cyberinfrastructure project.

The European Union's Horizon Europe research and innovation programme: 101059548.

## Competing Interests

The authors declare there are no competing interests.

## Author Contributions

- Carmelo Bonannella conceived and designed the experiments, performed the experiments, analyzed the data, prepared figures and/or tables, authored or reviewed drafts of the article, and approved the final draft.
- Tomislav Hengl conceived and designed the experiments, performed the experiments, analyzed the data, authored or reviewed drafts of the article, and approved the final draft.
- Leandro Parente performed the experiments, prepared figures and/or tables, authored or reviewed drafts of the article, and approved the final draft.
- Sytze de Bruin conceived and designed the experiments, authored or reviewed drafts of the article, and approved the final draft.

## Data Availability

The training dataset used for the study is publicly available at the University of Reading: https://researchdata.reading.ac.uk/99/

The environmental covariates used as predictive variables for the model is available at CHELSA and The University of Tokyo:

https://chelsa-climate.org/downloads/

http://hydro.iis.u-tokyo.ac.jp/~yamadai/MERIT_DEM/

The code for the model implementation is similar to a previous publication from the same authors and is available at GitLab: https://gitlab.com/geoharmonizer_inea/spatial-layers/-/blob/master/veg_tree.species_anv.pnv.eml/vegetation_mapping_functions.R

The outputs are available on Zenodo:

Bonannella, Carmelo, Hengl, Tomislav, Leal Parente, Leandro, & de Bruin, Sytze. (2023). Current and future global distribution of potential biomes under climate change scenarios (Version 2) [Data set]. Zenodo. https://doi.org/10.5281/zenodo.7822868.

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
