# Peer review of "Biomes of the world under climate change scenarios: increasing aridity and higher temperatures lead to significant shifts in natural vegetation"

_PeerJ, doi:10.7717/peerj.15593_

## Round 0.1 · original submission · Minor Revisions

Global climate changes affect the functioning, distribution and structure of natural vegetation, the preservation of which in the conditions of climate change is one of the priority scientific tasks. The study concerns the parameterization of the relationships between climate and vegetation, as well as the dependence and prediction of biomes under different climate change scenarios. The main thing is the application of modeling methods and machine learning algorithms. Scientific text is interesting to read and well structured, it deserves attention. Nevertheless, after a detailed review of the proposed materials, expert reviewers made a number of recommendations to improve the quality of the article. Therefore, please familiarize yourself with the reviewers' recommendations and, if possible, take them into account.

Reviewer 1 ·

Basic reporting

The authors use clear, unambiguous, professional English language throughout. The literature is well referenced & relevant. The structure conforms to standards. Figures are relevant, high quality, well labelled & described. The authors did not generate new data for the analyses.
What could be improved in the introduction is the description of the state-of-the-art of modelling biomes with statistical prediction models and a description of the deficiencies of existing works that warrant the present analysis. Some related studies are mentioned in the discussion, but I think they should also be mentioned in the introduction to explain how others have tackled the problem and how the present study represents an improvement.

Experimental design

The study presents original primary research within scope of the journal. The research question is well defined, relevant & meaningful. As described above it could be better described how the research fills an identified knowledge gap.
I do not see any technical mistakes in the statistical modelling. However, as the authors note, the training data has a strong spatial bias that is not accounted for in the analysis and poses as “serious limitation” (L464). Methods are described with sufficient detail & information to replicate.

Validity of the findings

I have limited confidence in the robustness of the findings and describe in my comments why. The conclusions are well stated, linked to original research question & limited to supporting the results.

Additional comments

The study projects the future distribution of global biomes (n=20) and identifies regions that will likely experience a biome transition. To this end, the authors use an ensemble machine learning model to parametrize biome-climate relationships using a dataset of circa 9000 training sites with known biome state and 66 climatic predictor variables (plus 6 topographic variables). The fitted ensemble model is then used to predict the biome state of global grid cells today and in two future time steps under three emission scenarios (RCP 2.6, 4.5 and 8.5) using climate projections from one General Circulation Model. Less than 1% of the global land surface was projected to likely experience a biome transition by 2070 (the mean year of the period covered by climatology 2061-2080). Overall, I thought that manuscript addresses an important and timely topic and was easy to understand. I could not identify any errors in the statistical modelling.

There are however some model and data deficiencies that challenge the rigour of the results. First and foremost, the prediction certainty for most biome classes is low (0.5 on average; see Table 1). This means that the model makes many wrong predictions over potentially large geographic areas. I would therefore assume that if you produced a global map of the biome classes based on your fitted ensemble model, the map would contain several implausible biome class predictions. Unfortunately, no such map is presented, but I think it should be presented as it is a useful sanity check. Crucially, if your model’s ability to reliably predict the biome class today is low, it can also not reliably predict the future biome class and biome transitions.

The limited prediction certainty may partly be due to spatial biases in the training dataset. As you acknowledge (L462ff), your training dataset is “heavily imbalanced” spatially and regarding the representation of the different biomes, and this poses a “serious limitation of the study, as the model may not be able to accurately predict the vegetation in those locations”. However, the spatial variation in prediction certainty is not quantified and visualized, which makes it difficult for the reader to assess how robust the predictions are in different regions after all. You could plot a map showing the maximum probability of observing a biome class (as e.g. in Boonman et al. 2022 and Lindgren et al. 2021). To improve prediction certainty, is it feasible to include more training sites in data-deficient regions by compiling more pollen sites?

Second, to fit the models, present-day (1979-2013) climate data was used to predict the biome distribution 6000 years ago. To me, this seems sensible only when the values of the climatic predictor variables were the same 6000 years ago as at present. Otherwise, you do not accurately parametrize the climate-vegetation relationships that you then use to predict present and future biome distributions. This further reduces the confidence in your results.
Can you explain the rationale behind the choice to model past vegetation states with present-day climate data? I would not really be convinced by a response that says “the global mean temperature was similar to the present and therefore, our approach is defendable” since the global mean may not be informative for some regions, and you also used 65 other climatic predictor variables. You would have to show that at each of the training sites, the values of all your 66 climatic predictor variables would be the same 6000 years ago as they are today, which is probably not possible. Alternatively, you would have to demonstrate that the sites are still in the same biome state as they were 6000 years ago.

Third, you do not account for extrapolation uncertainty. Meyer & Pebesma (2021) showed that machine learning algorithms cannot reliably predict outside the training data domain (especially when the training data is highly clustered, as in your case). I would like to see where you make future projections into novel climatic conditions in terms of novel data ranges and combinations of climate variables. Meyer & Pebesma (2021) provide a method (and R code) to estimate the area of applicability of spatial prediction models such as yours. I think it is mandatory to check where your model is not applicable under the future climate scenarios.
In the discussion (L379) you write that “ensemble models reduce model uncertainty and are more robust towards extrapolation”. While I agree that ensemble models can reduce prediction uncertainty in the training data domain, this does not necessarily mean they extrapolate better. The two studies you cite in support of this claim (Mehra et al. 2019, Bonannella et al. 2022) do not show that ensembles extrapolate better.

Also, note that the bioclimatic variables you use have limited transferability themselves. Specifically, synthetic variables like “precipitation of warmest quarter” or “precipitation of driest quarter”. The reason is that the warmest or driest quarter in the future may not be the same three months as today. Shifts in the timing of rainfall and warmth availability matter for organisms, but the synthetic bioclimatic variables do not capture shifts in this timing. This is a problem that cannot be resolved by the method of Meyer & Pebesma (2021). A solution could be to exclude the bioclimatic predictor variables.


Minor comments
Unlike what was written in last line of the abstract, maps of prediction error are not provided.

I thought that in the introduction you could improve the description of the state-of-the-art of modelling biomes with statistical prediction models and a description of the deficiencies of existing works that warrant your analysis. Some related studies are mentioned in the discussion, but I think they should also be mentioned in the introduction to explain how others have tackled the problem and in which ways your study represents an improvement.

You only used future climate projections from one GCM (MPI-ESM-mr). It is recommended to use at least five GCMs to represent the uncertainty in ecological projections originating from uncertainty in climate trajectories (see recommendations in https://chelsa-climate.org/future/) .

In the results section, please use the names of the biomes rather than Tx and Ty.

Lines 487ff stress that the margin of victory prevents us from making incorrect conclusions, but I was wondering how informative the margin of victory is when the probabilities of individual biome classes cannot reliably be estimated in the first place?

You stressed several times (e.g. L504) that you used expert knowledge to select only a limited number of predictor variables, but after all you used all available climatic predictors in CHELSA 1.2. So I would not agree with both that you used expert knowledge to select and that you used a limited number of predictor variables


References
Bonannella, C., Hengl, T., Heisig, J., Parente, L., Wright, M. N., Herold, M., and de Bruin, S. (2022). Forest tree species distribution for Europe 2000–2020: mapping potential and realized distributions using spatiotemporal machine learning. PeerJ.
Boonman, C. C. F., Huijbregts, M. A. J., Benítez-López, A., Schipper, A. M., Thuiller, W., & Santini, L. (2022). Trait-based projections of climate change effects on global biome distributions. Diversity and Distributions, 28, 25– 37. https://doi.org/10.1111/ddi.13431
Lindgren, A., Lu, Z., Zhang, Q., & Hugelius, G. (2021). Reconstructing past global vegetation with random forest machine learning, sacrificing the dynamic response for robust results. Journal of Advances in Modeling Earth Systems, 13, e2020MS002200. https://doi.org/10.1029/2020MS002200
Mehra, A., Tripathy, P., Faridi, A., and Chinmay, A. (2019). Ensemble learning approach to improve existing models. International Journal of Innovative Science and Research Technology, 4.
Meyer, H., & Pebesma, E. (2021). Predicting into unknown space? Estimating the area of applicability of spatial prediction models. Methods in Ecology and Evolution, 12, 1620– 1633. https://doi.org/10.1111/2041-210X.13650

·

Basic reporting

The manuscript is well-written and easy to read and the figures are beautifully presented

Experimental design

The methodology is sound and well-described. The paper is an important contribution as it improves on a number of aspects of previous research, such as the approach taken to validation and model ensembling.

Validity of the findings

There are a few places where I would like to see the authors elaborate and places where I think the authors have presented an overly simplistic overview of existing approaches. Furthermore, the divergence in the results presented here from those often produced from process-based models and trends observed in situ and from satellites needs to be directly addressed. There is no mention of the CO2 greening effect and how this impacts (or rather doesn’t impact) model predictions. Lastly, some of the language used to describe forest degradation needs to be polished to avoid adding to the confusion around savanna being a form of degraded forest and upsetting savanna and grassland ecologists who have been battling this misconception for decades. These points are elaborated upon in the additional comments

Additional comments

The paper overly simplifies and creates some false dichotomies between types of models. In particular, rather than a dichotomy, there is a spectrum of models from process-based to data-based. For example, Higgins et al. 2012 (Higgins, Steven I., et al. "A physiological analogy of the niche for projecting the potential distribution of plants." Journal of Biogeography 39.12 (2012): 2132-2145.) present a model that is data-driven but includes a simple representation of process parameterized from data. Likewise, most dynamic vegetation models have a number of tuned parameters that are calibrated to ensure that models match data (though they often don’t advertise this). The authors should acknowledge the continuum.

The introduction seems to perpetuate what I believe to be a false narrative that there is some ‘true’ or ‘consistent’ notion of a biome. The authors acknowledge that some biomes schemes are subjective( L86-L88). I believe that all biome schemes are subjective to a degree. Whether you chose to use a biome scheme based on structural vegetation properties or floristics depends on the process you are hoping to describe or understand. The papers by Higgins et al 2016 and Moncrieff et al 2015 cited in the text include some relevant discussion of this. I would like to see a little more discussion of this topic, in particular a more explicit identification of why the biome scheme used is appropriate for the task at hand.


Including (explicitly or implicitly) climate in the definition of biomes needs to be further addressed. If the purpose of the study is to investigate how climate will change biome distributions this will introduce a degree of circularity. More concretely, let's say we are interested in a location where the biome is currently mapped as cold deciduous forest, and that this biome is defined (among other things) as having a mean annual temperature of <10 degrees C. If the climate warms to 12 C, but nothing functionally, structurally, or floristically changes, we would still say a biome shift has occurred. This seems incorrect to me. Even if climate is not explicitly included in the definition (as is my understanding in the BIOME600 Data used here), calling a biome 'cold ...' implicitly biases us to define this biome in terms of its current climate.

L421-448. The use of the word 'savannization' to describe the degradation of tropical forest is very contentious. Grassland and savanna ecologists strongly object to this. These ecosystems are incredibly diverse and valuable, and bear no resemblance to degraded forests. I think it's fine to use this term to describe the natural (or climate-driven) transitions between forest and more open ecosystems, but you should avoid using it if you are referring to forest degradation.

The CO2 fertilization effect: One of the major reasons for using process-based models to investigate vegetation shifts is their ability to incorporate the effect of increasing atmospheric CO2 concentrations. This is not included here understandably, as it is not possible to parameterize the CO2 effect when CO2 is spatially uniform (mostly). The effects of not including this factor in the model need to be discussed. This is particularly important as most process-based models predict a large greening effect in the tropics as a result of increasing CO2. This is the opposite of the browning or savannization predicted here. The fact that data-driven models generally predict this greening in the tropics, and process-based models generally predict browning, leaves us very uncertain about the future biome distribution in the tropics. This is a key finding in the IPCC AR6 chapter on Africa. It must also be acknowledged that a widespread pattern of greening has been observed from satellite-based and in-situ studies across the tropics. E.g.,
Anchang, Julius Y., et al. "Trends in woody and herbaceous vegetation in the savannas of West Africa." Remote Sensing 11.5 (2019): 576.
Stevens, Nicola, et al. "Savanna woody encroachment is widespread across three continents." Global Change Biology 23.1 (2017): 235-244.

This provides evidence supporting the greening predictions from process-based models. The authors should note and discuss this.

---

## Round 0.2 · accepted · Accept

The current issue of climate change and its impact on vegetation is the subject of research in this manuscript. After re-parsing, the quality of the manuscript has improved. The obtained research results are relevant and verified, and therefore will be interesting for scientists and a wide range of readers.

Reviewer 1 ·

Basic reporting

no comment

Experimental design

no comment

Validity of the findings

no comment

Additional comments

I thank the authors for their swift but thorough revision and the inclusion of climate projections from additional GCMs. My concerns have largely been clarified and I can recommend this study for publication.

A few very minor comments: The authors responded to my comments on model transferability that they could not implement methods to delimit the extrapolation space. This is fine, but in my opinion, extrapolation uncertainty should be mentioned as a remaining uncertainty of this study. Similarly, the authors responded to my comment on the BIOCLIM variables that they are frequently used in SDM studies that make future projections. That other studies make the same invalid assumption on stationarity of these variables is not an excuse for making the same invalid assumption. But I found their response that the synthetic “quarter” variables were not the most important in this study convincing. So why not mention the limited transferability of the synthetic BIOCLIM variables and then make this argument? I think these points would make the discussion more balanced and stronger. But again, this is only a very minor suggestion and I leave this up to the authors.